# Molecular architecture of human polycomb repressive complex 2

**Claudio Ciferri[1]\*[†], Gabriel C Lander[2], Alessio Maiolica[3], Franz Herzog[3], Ruedi Aebersold[3,4], Eva Nogales[1,2,5]\***

[1]Department of Molecular and Cell Biology, University of California, Berkeley, United States; [2]Life Sciences Division, Lawrence Berkeley National Laboratory, Berkeley, United States; [3]Department of Biology, Institute of Molecular Systems Biology, ETH Zurich, Zurich, Switzerland; [4]Faculty of Science, University of Zurich, Zurich, Switzerland; [5]Howard Hughes Medical Institute, UC Berkeley, Berkeley, United States

**Abstract** Polycomb Repressive Complex 2 (PRC2) is essential for gene silencing, establishing transcriptional repression of specific genes by tri-methylating Lysine 27 of histone H3, a process mediated by cofactors such as AEBP2. In spite of its biological importance, little is known about PRC2 architecture and subunit organization. Here, we present the first three-dimensional electron microscopy structure of the human PRC2 complex bound to its cofactor AEBP2. Using a novel internal protein tagging-method, in combination with isotopic chemical cross-linking and mass spectrometry, we have localized all the PRC2 subunits and their functional domains and generated a detailed map of interactions. The position and stabilization effect of AEBP2 suggests an allosteric role of this cofactor in regulating gene silencing. Regions in PRC2 that interact with modified histone tails are localized near the methyltransferase site, suggesting a molecular mechanism for the chromatin-based regulation of PRC2 activity.

**\*For correspondence:** enogales@lbl.gov (EN), claudio.ciferri@novartis.com (CC)

[†]**Present address:** Novartis Vaccines and Diagnostics, Cambridge, United States

**Competing interests:** The authors have declared that no competing interests exist

## Introduction

The combined antagonistic activities of the Polycomb group (PcG) and the Trithorax group (TrxG) protein complexes contribute to correct homeotic gene expression and accurate cell fate maintenance. PcG and TrxG complexes are evolutionarily conserved in eukaryotes and regulate chromatin structure through recruitment to Polycomb or Trithorax Responsive Elements (PREs, TREs) for gene silencing or activation, respectively (*Schuettengruber et al., 2007*). Within the PcG protein complexes, Polycomb Repressive Complex 2 (PRC2) plays an essential role in chromatin modification by di- and tri-methylating Lys 27 of Histone H3 (H3K27me2/3) (*Cao et al., 2002*; *Czermin et al., 2002*; *Muller et al., 2002*; *Kirmizis et al., 2004*). In higher eukaryotes, high levels of H3K27me3 are typically associated with transcriptional repression and gene silencing. The importance of the PcG protein in this process is emphasized by the observation that deletion of any of its components in mice results in early embryonic lethality or severe defects during development (*O'Carroll et al., 2001*; *Pasini et al., 2004*).

The PRC2 core, conserved from *Drosophila* to humans, is composed of four proteins that add up to about 230 kDa (*Figure 1A*) (see *Margueron and Reinberg, 2010* for a recent review): EED (present in different isoforms), either one of the two methyltranferases Ezh1 or Ezh2 (Ezh1/2), Suz12, and either RbAp46 or RbAp48 (RbAp46/48). Both Ezh1 and Ezh2 contain enzymatic methyl-transferase activity within a C-terminal SET (Su(var)3-9, Enhancer-of-zeste, Trithorax) domain. PRC2 complexes containing Ezh1 have lower enzymatic activity than those containing Ezh2, and target a subset of Ezh2 genes (*Margueron et al., 2008*). Ezh2 activity appears to depend on interaction with both Suz12 and the WD40 domain in EED (*Cao and Zhang, 2004a*; *Pasini et al., 2004*; *Yamamoto et al., 2004*; *Ketel*

**eLife digest** Protein complexes—stable structures that contain two or more proteins—have an important role in the biochemical processes that are associated with the expression of genes. Some help to silence genes, whereas others are involved in the activation of genes. The importance of such complexes is emphasized by the fact that mice die as embryos, or are born with serious defects, if they do not possess the protein complex known as Polycomb Repressive Complex 2, or PRC2 for short.

It is known that the core of this complex, which is found in species that range from *Drosophila* to humans, is composed of four different proteins, and that the structures of two of these have been determined with atomic precision. It is also known that PRC2 requires a particular protein co-factor (called AEBP2) to perform this function. Moreover, it has been established that PRC2 silences genes by adding two or three methyl (CH3) groups to a particular amino acid (Lysine 27) in one of the proteins (histone H3) that DNA strands wrap around in the nucleus of cells. However, despite its biological importance, little is known about the detailed architecture of PRC2.

Ciferri et al. shed new light on the structure of this complex by using electron microscopy to produce the first three-dimensional image of the human PRC2 complex bound to its cofactor. By incorporating various protein tags into the co-factor and the four subunits of the PRC2, and by employing mass spectrometry and other techniques, Ciferri et al. were able to identify 60 or so interaction sites within the PRC2-cofactor system, and to determine their locations within the overall structure.

The results show that the cofactor stabilizes the architecture of the complex by binding to it at a central hinge point. In particular, the protein domains within the PRC2 that interact with the histone markers are close to the site that transfer the methyl groups, which helps to explain how the gene silencing activity of the PRC2 complex is regulated. The results should pave the way to a more complete understanding of how PRC2 and its cofactor are able to silence genes.

*et al., 2005*). EED's WD40 beta propeller, in turn, interacts with H3K27me3 repressive marks. This interaction is proposed to promote the allosteric activation of PRC2 methyltransferase activity (*Margueron et al., 2009*; *Xu et al., 2010*). RbAp48 also contains a WD40 propeller required for interaction with both Suz12 and the first 10 residues of unmodified Histone H3 peptides (*Nowak et al., 2011*; *Schmitges et al., 2011*).

Biochemical studies have shown that PRC2 co-purifies with the protein AEBP2 and it has been proposed that this interaction aids in targeting of PRC2 to specific DNA sites and enhances its methyl-transferase activity (*Cao and Zhang, 2004b*; *Kim et al., 2009*). This important cofactor is an evolutionarily conserved protein present in two isoforms in humans, an adult-specific larger form (51 kDa) and an embryo-specific smaller form (32 kDa), both containing three Gli-Krüppel (Cys2-His2)-type zinc fingers (*He et al., 1999*).

The PRC2 complex has been the focus of a significant number of biochemical and molecular studies (for a recent review see *Margueron and Reinberg, 2011*), and atomic structures of the EED and RbAp48 subunits have been reported (*Margueron et al., 2009*; *Nowak et al., 2011*; *Schmitges et al., 2011*; *Xu et al., 2010*). A comprehensive picture, however, of the PRC2 complex and the manner in which its different components interact to coordinate the regulated methyl-transferase activity remains elusive. In this study we present the first structure of the entire PRC2 holoenzyme in complex with AEBP2, describing its subunit architecture and the important interactions between their domains. These results contribute significantly to our structural understanding of this chromatin regulator, allowing us to suggest a possible molecular mechanism for PRC2's role in gene silencing.

## Results

### Reconstitution of the human PRC2-AEBP2 complex

In preliminary experiments, we reconstituted the tetrameric PRC2 complex (Ezh2/EED/Suz12/RbAp48) in an insect cell expression system, following previous protocols established to reconstitute a functional PRC2 complex (*Pasini et al., 2004*; *Ketel et al., 2005*; *Margueron et al., 2008*). When analyzed

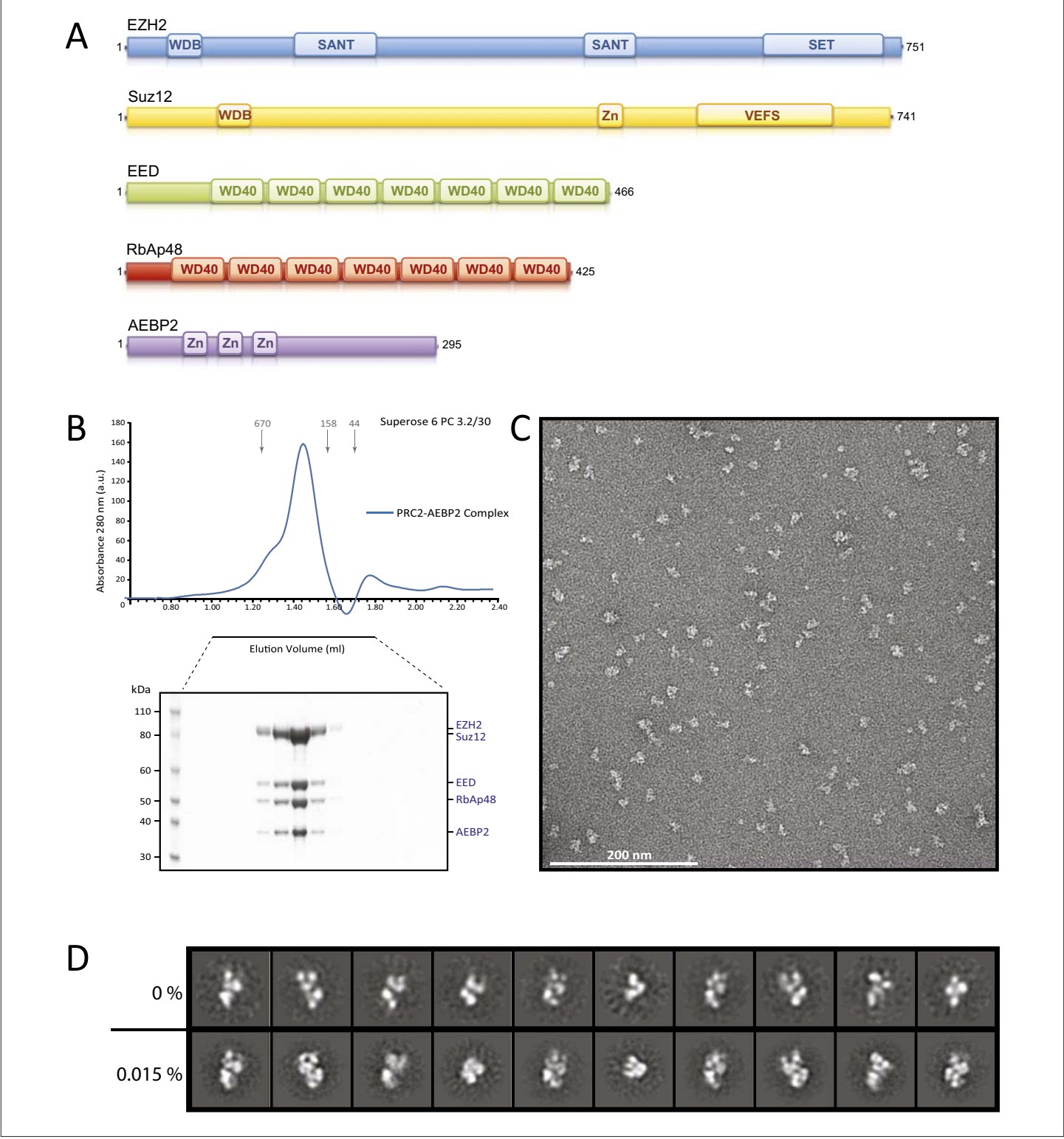

**Figure 1**. Reconstitution of the human PRC2-AEBP2 Complex. (**A**) Schematic representation of the components of the human PRC2 Complex and AEBP2. (**B**) Size-exclusion chromatography of recombinant PRC2-AEBP2 complex and corresponding SDS-PAGE separation stained with Coomassie brilliant blue. The molecular mass of the recombinant complex is ~275 kDa. Arrows and numbers indicate elution markers in the size-exclusion chromatography experiments and their molecular masses (in kilodaltons), respectively (a.u.: arbitrary unit). (**C**) Negative-stain EM of the recombinant PRC2-AEBP2 complex. Individual particles have an elongated shape with a length of ~16 nm and a thickness of ~7 nm. Bar: 200 nm. (**D**) Comparison between representative reference-free 2D class averages from non cross-linked (0%) and mildly cross-linked (0.015% glutaraldehyde) particles of the PRC2-AEBP2 complex.

by SDS page the purified complex appeared to be stoichiometric and biochemically homogeneous (data not shown). However, further analysis to generate reference-free 2D class averages and a 3D reconstruction were limited in resolution and lacked clear structural details (data not shown), indicating the presence of extreme conformational flexibility and hampering further structural studies of this tetrameric PRC2.

Since the AEBP2 cofactor is known to stably interact with the PRC2 complex and is required for optimal enzymatic activity (*Cao and Zhang, 2004b*), we added AEBP2 to the expression system, with the hope of locking the PRC2 complex in a conformation, both more active and more stable. We over-expressed the five-component PRC2-AEBP2 complex using the shorter isoform of AEBP2 protein, which is missing an extended unstructured region present in the longest isoform. Stoichiometric and biochemically homogeneous PRC2-AEPB2 complex was purified from insect cell lysate by affinity chromatography using a StrepII tag on the AEBP2 subunit, followed by size exclusion chromatography (*Figure 1B*). The subunits of the purified complex appeared to be stoichiometric by SDS page, although we could not rule out the presence of a small excess of AEBP2, as this subunit was utilized for tag-based purification. In our EM images we observed a small percentage of very small particles, which could be attributed to views of the complex oriented perpendicular to the plane of the grid, or to the presence of partial complexes (even AEBP2 by itself). These smaller particles were not considered in the generation of the 3D reconstruction (see 'Materials and methods'). While inclusion of the cofactor significantly improved the quality of the 2D class averages in EM analysis (*Figure 1D*, top row), both detailed inspection and attempts to pursue a 3D reconstruction (not shown) indicated that the stability of the complex remained limiting. In order to preserve complex stability for further structural analysis, we mildly cross-linked the PRC2-AEBP2 complexes using 0.015% glutaraldehyde. Under these conditions we saw no apparent aggregation of the complex (*Figure 1C*). Importantly, comparison of class averages of the non cross-linked and the cross-linked sample (*Figure 1D*, bottom row), shows that the cross-linking had a positive effect on the stability of the complex, as judged by the increased detail in the class averages (for similar number of particles and classes). Class averages show PRC2 having an elongated shape, with top and bottom density regions that are joined at a narrow central point (*Figure 1D*).

## Architecture of the PRC2-AEBP2 complex

Given the lack of structural information for the PRC2 complex, we carried out ab initio structural determination using the Random Conical Tilt (RCT) method (*Radermacher et al., 1987*) (*Figure 2*). Tilted and untilted image pairs were collected (*Figure 2A*), and 20 reference-free 2D class averages were calculated from the untilted classes (*Figure 2B*). RCT reconstructions were generated for each class average using the corresponding tilted images (*Figure 2B,C*). A four-lobed organization was evident in all the resulting models, indicating that the complex likely assumes one major architectural state. Concurrently, we collected a large dataset (~40,000 particles) of untilted images, which were used to generate 1000 reference-free 2D class averages. Euler angles were assigned to these class averages, using each of the 20 RCT reconstructions as reference models. The results showed that the class averages were aligned in 3D in a self-consistent fashion, producing a similar complex architecture at the low resolution expected at this stage, irrespective of the RCT structure used as a reference (data not shown). Not surprisingly, Euler assignment using the RCT model from the most populated class gave a reconstruction that had the most structural detail (*Figure 2C*, class 4). This reconstruction was then used as an initial model (*Figure 3A*) for multiple rounds of projection-based angular refinement for single particles of the full 0° dataset (*Penczek et al., 1994*). The final refined structure has a resolution of 21 Å, based on the 0.5 Fourier shell correlation criteria (*Figure 3B*) (*van Heel and Schatz, 2005*). Due to the shape of the complex and its tendency to lie flat on the grid, we do not have an even distribution of views. However, views around the longitudinal axis of the model (vertical axis in the Euler plot) are well represented (*Figure 3C*). The quality of our model in representing the experimental images was confirmed by comparing the re-projected 3D reconstruction of the PRC2-AEBP2 complex with reference-free class averages (*Figure 3D*).

*Figure 4* shows four orthogonal views of the PCR2-AEBP2 structure, displayed as an isosurface that corresponds to the estimated molecular mass for the complex (275 kDa). The structure (roughly 160 Å by 120 Å by 90 Å) consists of four large lobes: A, B, C and D (each approximately 55 Å in diameter), interconnected by two narrower arms, Arm 1 at the top, and Arm 2 in the center. In the upper part of the structure, Arm 1 spans the width of the complex horizontally, connecting lobes A and B. These

Biophysics and structural biology

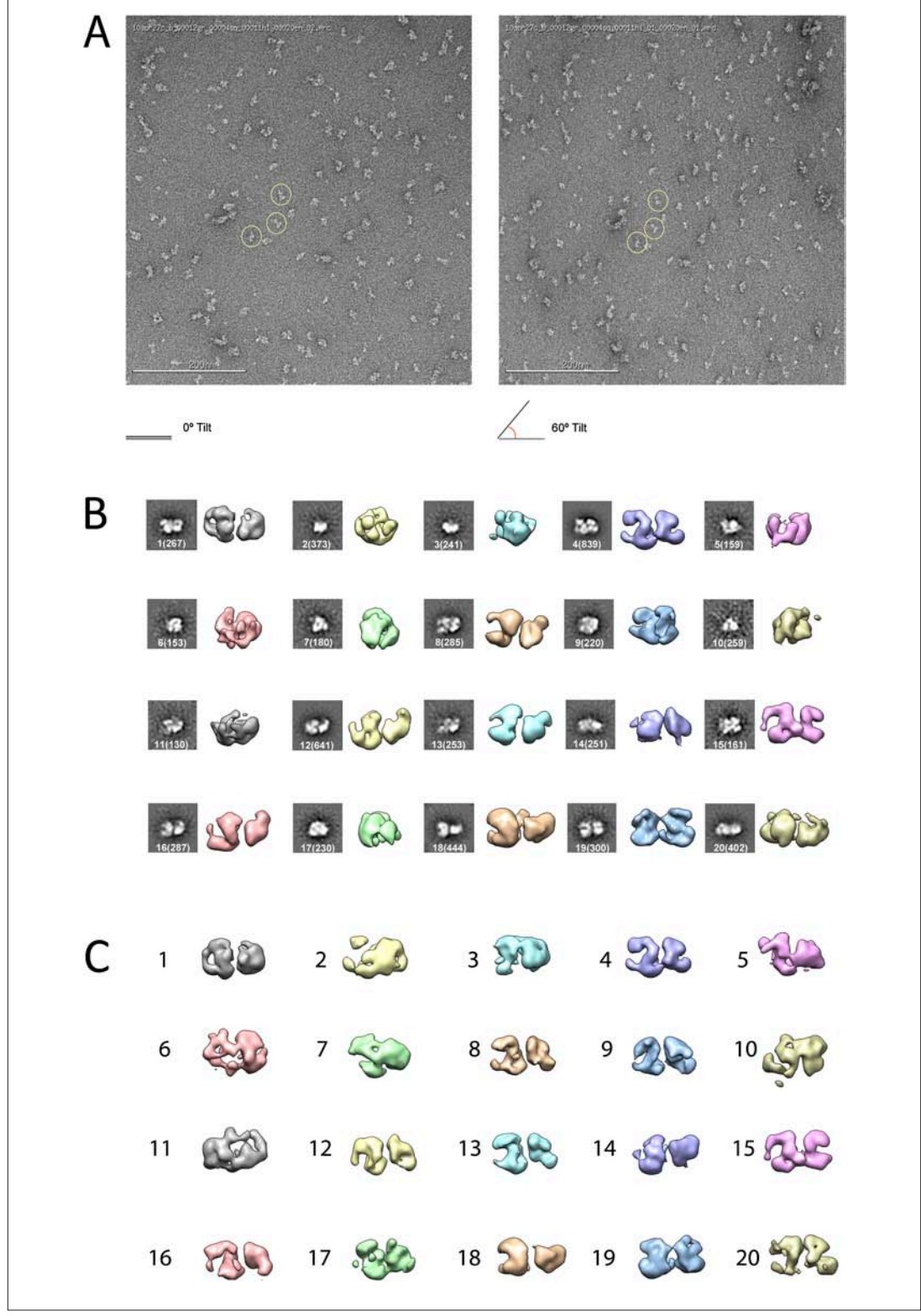

Figure 2. Ab initio random conical tilt reconstruction of the human PRC2-AEBP2 complex. (A) Representative untilted and 60° tilt-pair micrographs (80,000× magnification). Corresponding particles pairs indicated by yellow circles. (B) RCT Volumes aligned to each of the 20 corresponding reference free class averages (from 6075 particles in the 0° micrographs, each class containing between 150 and 800 particles, as indicated in parentheses). (C) Alignment of the RCT volumes with respect to each other.

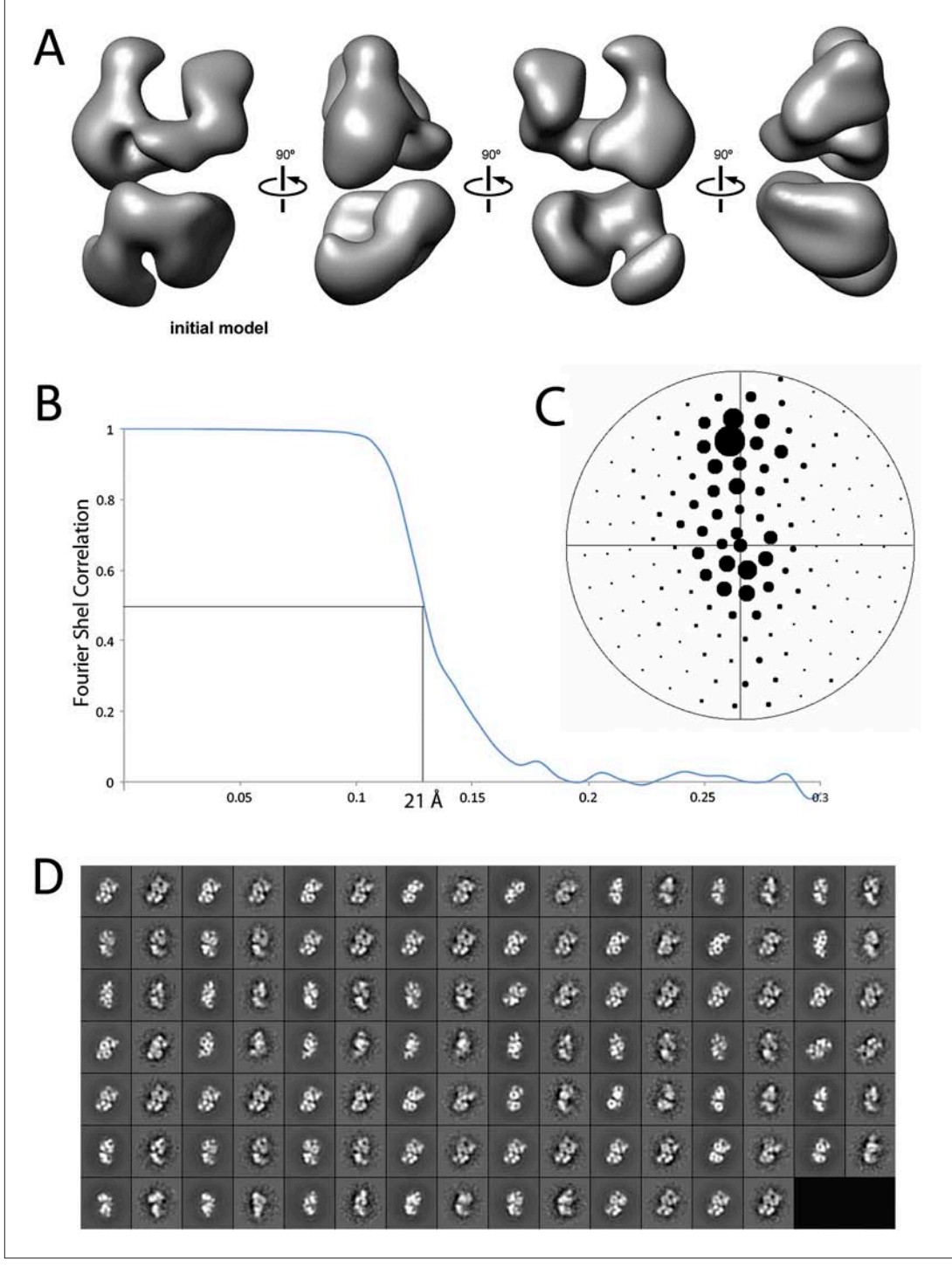

**Figure 3**. Refinement and final statistics for the reconstruction of the human PRC2-AEBP2 complex. (**A**) 3D reconstruction of the ab initio model used for iterative projection-matching (corresponds to reconstruction 4 in *Figure 2C*). (**B**) Final resolution was estimated as 21 Å using the Fourier shell correlation criterion with a cutoff of 0.5. (**C**) Euler distribution plot of the PRC2-AEBP2 particles, where the size of the circle corresponds to the relative number of views included for that projection. (**D**) Comparison between re-projection of the final model (even numbers) and reference free 2D class averages (odd numbers). The projections were generated at 12° increments around the PRC2 long axis, including out-of-plane tilting going up to 30°.

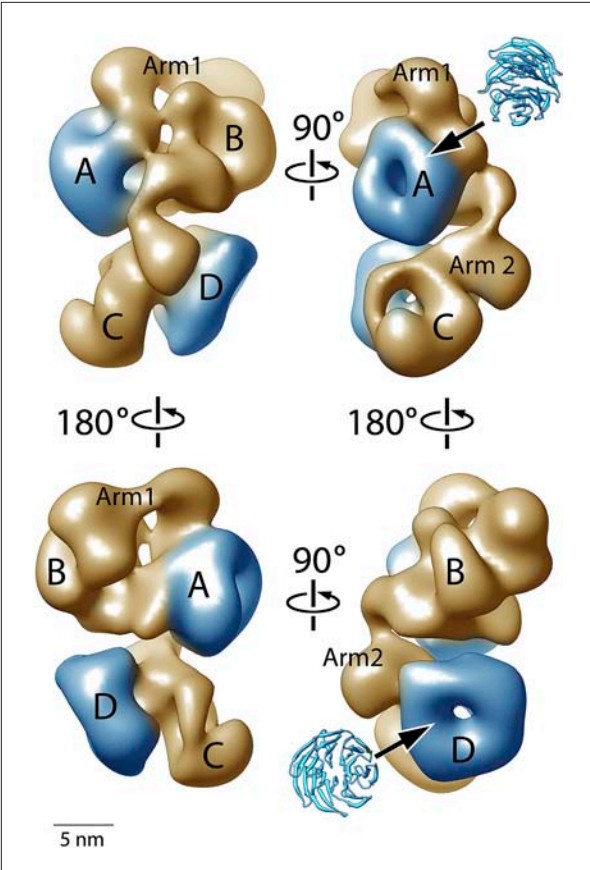

**Figure 4**. Structure of the human PRC2-AEBP2 complex. (**A**) The PRC2-AEBP2 complex consists of 4 different lobes, each about 55 Å in diameter (A, B, C, D), interconnected by two narrow arms (Arm 1, Arm 2). The two WD40 domains of EED and RbAp48 (indicated in blue) are located at opposite ends. DOI: 10.7554/eLife.00005.006

lobes have additional interactions closer to the center of the particle. Arm 2 runs in a more vertical direction, starting at Lobe A and continuing vertically to the lower half of the structure, where it merges with Lobe C and D. Lobe A and D both have distinct ring-like shapes that are consistent with the propeller-like WD40 motifs of EED and RbAp48 (*Figure 4*, blue). These two WD40 domains are the only regions in the PRC2 complex for which atomic structures are presently available.

## Interaction mapping using chemical cross-linking and mass spectrometry

To identify regions of interaction within the PRC2-AEBP2 complex, we carried out chemical cross-linking using isotope labeled cross-linkers and mass spectrometric identification of the cross-linked peptides, on recombinant complexes identical to those used for structure determination. This technique has been successfully utilized in several laboratories during recent years to aid in characterizing the organization of multi-subunit protein complexes (*Gingras et al., 2007*; *Maiolica et al., 2007*; *Ciferri et al., 2008*; *Rappsilber, 2011*). For cross-linking analysis, we used DSSG, a homo-bifunctional cross-linker reacting with primary amines in lysine side chains or protein N-termini spaced ~7 Å apart. We identified about 60 intra- or inter-molecular sites of interaction within the PRC2-AEBP2 complex (*Figure 5*; *Supplementary file 1*).

While some of these interactions confirm previous biochemical data, others had never been described before and thus significantly expanded our knowledge of the PRC2 organization. In this report we include all the identified cross-linking interactions (*Supplementary file 1*). None of them proved in contradictions with previous reports or published crystal structures. Salient cross-linked Lysines are described in the following section, indicated as [$K_n$], where n represents the residue number of the protein.

Previous crystallographic analysis of the EED-Ezh2 complex described the interaction of the C-terminal WD40 propeller of EED with residues 40–68 of Ezh2 (*Han et al., 2007*). In agreement with these data, we identified a cross-link between EED [$K_{423}$] and Ezh2 [$K_{39}$], as well as an internal cross-link between EED [$K_{79/83}$] and EED [$K_{433}$] (*Figures 5 and 6A*). We found two additional interactions between EED and other PRC2 components (*Figure 5*), one connecting the WD40 domain of EED [$K_{433}$] with a loop between the second and the third Zn fingers of AEBP2 [$K_{118/121}$], and the other linking the N-terminal region of EED [$K_{19/20}$] with a region downstream of the first SANT domain of Ezh2 [$K_{241/243}$] (*Figure 5*).

We found several interactions involving this first SANT domain of Ezh2. The peptide centered at residues [$K_{210/217/222}$] binds the second SANT domain of Ezh2 [$K_{510/514/515}$], associates with the VEFS domain of Suz12 [$K_{650}$], and is also cross-linked to the same region of AEBP2 [$K_{118/121}$] that interacts with EED. Interestingly, this region of AEBP2, which contains the last two Zn finger domains [$K_{95}$–$K_{166}$], represents an intricate hub of interaction within the PRC2 complex. In addition to the interactions described above, cross-linking data show that AEBP2 binds to Ezh2's methyltransferase (SET)-containing C-terminus [$K_{568}$–$K_{740}$], as well as Suz12's Zn finger domain [$K_{414}$–$K_{534}$]. Furthermore, this

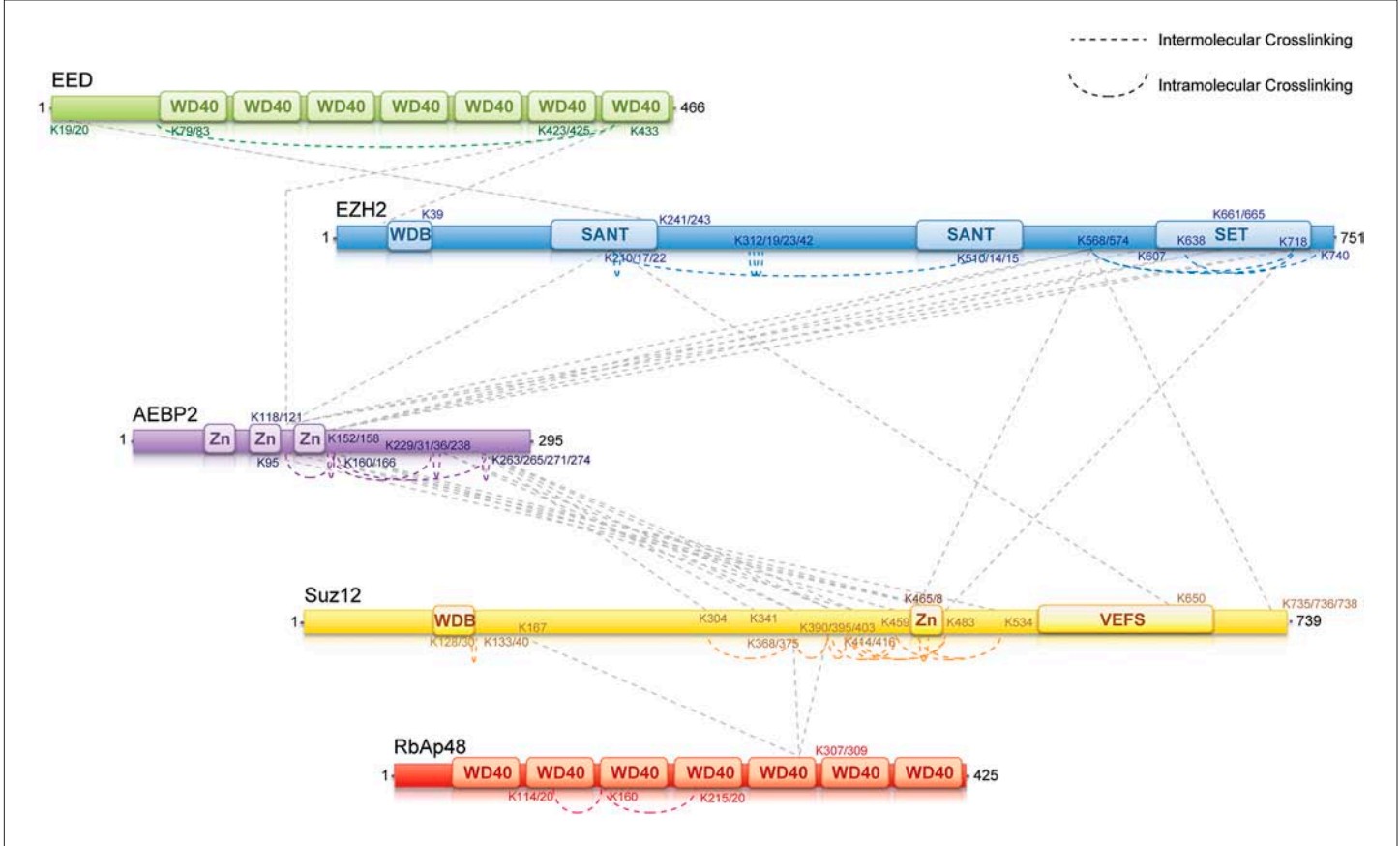

**Figure 5**. MS-coupled cross-linking analysis of the PRC2-AEBP2 complex: Cross-link map for PRC2 in complex with AEBP2. Observed inter-molecular cross-links (straight dashed lines) are colored in grey. Intra-molecular cross-links are color coded by the respective PRC2-AEBP2 subunit (See also *Supplementary file 1*).

region of Suz12 [$K_{465/468}$–$K_{483}$] also interacts with the methyltransferase domain of Ezh2 [$K_{568}$–$K_{740}$], suggesting a trimeric subcomplex that includes the SET domain of Ezh2 and the Zn fingers of Suz12 and AEBP2. Based on the observations that the Zn finger of AEBP2 and the C-terminal region of Suz12 together interact with both the N-terminal SANT and C-terminal SET domains of Ezh2, we propose that these two terminal regions of Ezh2 are within proximity to, or directly interacting with each other.

Two recently published crystal structures of the *Drosophila* Nurf55-Suz12 complex identified a small region of interaction between the WD40 domain of Nurf55 (an ortholog of RbAp48) and an alpha helix of Suz12 (residues 77–93, corresponding to 105–121 in human) (*Nowak et al., 2011*; *Schmitges et al., 2011*). In addition, Schmitges and colleagues identified a region downstream of this Suz12 helix that plays an important role in stabilizing this interaction. Due to the absence of exposed lysines, we did not observe any cross-linking between the regions of RbAp48 and Suz12 shown to interact in the crystal structure. We did identify internal cross-linking within the WD40 propeller of RbAp48 [$K_{114/120}$–$K_{160}$–$K_{215/220}$], consistent with the reported crystal structure of the isoform RbAp46 (PDB: 3CFS; *Figure 6B*). Additionally, we detect the presence of a set of three interactions between the WD40 domain of RbAp48 [$K_{307/309}$] and the central portion of Suz12 [$K_{167}$–$K_{403}$] (*Figure 5*). This result suggests that the stabilizing downstream region described by Schmitges and colleagues might actually involve a larger portion of Suz12. In addition to this interaction with RbAp48, we also observe cross-linking between this region of Suz12 [$K_{304}$–$K_{403}$] and AEBP2 [$K_{229}$–$K_{274}$]. These residues of AEBP2 also interact with AEBP2's own Zn finger domain [$K_{95}$–$K_{166}$] (*Figure 5*). Taken together, these findings suggest that RbAp48, the central domain of Suz12, and AEBP2 undergo extensive interactions, in agreement with previous biochemical data (*Cao and Zhang, 2004b*).

In summary, our cross-linking data provides a detailed overview of the interactions among the components of the PRC2-AEBP2 complex. Interestingly, we find that AEBP2 plays a major role in

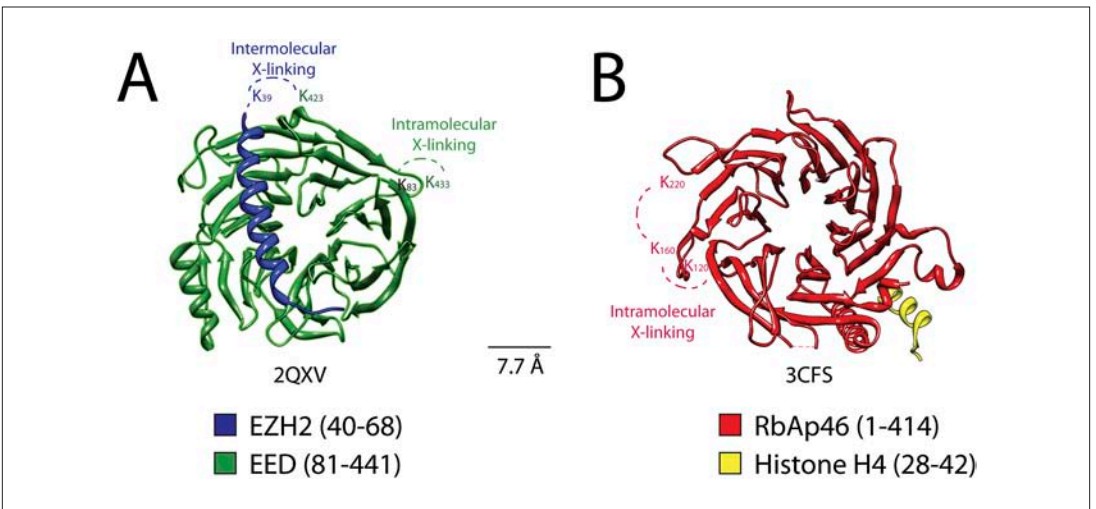

**Figure 6**. Isotopic crosslinking within the WD40 regions of EED and RbAp48. (**A**) Crystal structure of EED in complex with Ezh2 (PDB: 2QXV). EED is colored in green and Ezh2 in blue. Both intermolecular and intra-molecular crosslinking data fit very well with the inter-residue distance in the structure. Scale bar of 7.7 Å represent the length of DSSG molecule used for the cross-linking experiment. (**B**) Structure of the subunit RbAp46 (PDB: 3CFS; 89% identity to RbAp48). Distance between residues $K_{120}$ and $K_{160}$ (~5 Å) is within the range of the cross-linker. The larger distance observed between cross-linked residues $K_{160}$ and $K_{220}$ (~10 Å) could be justified if the length of both lysine residues is also included in the calculation.

coordinating different PRC2 subunits, explaining its stabilizing effect on the PRC2 complex. AEBP2's central region is involved in interactions with the N- and C-termini of Ezh2, EED, and the C-terminal region of Suz12, while its C-terminal region interacts with both the central portion of Suz12 and RbAp48.

## Subunit localization within the PRC2-AEBP2 complex

In order to provide a better-defined spatial context to all our cross-linking data we used EM analysis to localize the different PRC2 subunits within the EM structure of the AEBP2-bound complex. To determine the position of individual components, we incorporated an N-terminal MBP (maltose-binding protein) or an internal or C-terminal GFP (green fluorescence protein) into each of the subunits within the PRC2-AEBP2 complex, visualizing the resulting assembly by EM. We used reference-free 2D classification to identify the position of the specific tags when compared to the corresponding class averages of the untagged complex (*Figures 7–10*). We concentrated our analysis on a specific view that we define as 'canonical' (corresponding to the first panel in *Figure 4*), which is both well represented in the EM images (for most, although not all of the tagged samples) and where the structural features of the complex are clear and well separated.

We started by assigning the position of EED, which consists of a WD40 domain and an 80-residue N-terminus that is predicted to be unstructured (*Han et al., 2007*). Neither N-terminal MBP nor C-terminal GFP fusions assembled into stoichiometric or stable complexes. To overcome this limitation, we took advantage of the fact that the GFP N- and C-termini are in close spatial proximity and designed an expression vector to generate protein chimeras containing an internal GFP, connected by a short loop, at desired sites along the main protein chain. Using the crystal structure of EED (PDB: 2QXV), we designed four constructs, each containing an internal GFP at a distinct site within the WD40 domain. The two constructs containing an internal GFP at either residue 115 or 370 (EED-GFP$_{115}$ and EED-GFP$_{370}$) assembled into stoichiometric and stable complexes that were used for EM analysis (*Figure 7A*, panels I and II, respectively). Comparing the internally GFP-tagged EED sample with the untagged form allowed us to identify the additional GFP density protruding from the WD40-shaped Lobe A (*Figure 7A*). To further confirm this localization, we reconstituted and analyzed a complex containing an N-terminal MBP on Ezh2. According to the known crystal structure (*Han et al., 2007*), the EED WD40 domain interacts with the N-terminal region of Ezh2. Therefore, an MBP in this region should localize similarly to Lobe A. Consistent with this prediction, 2D analysis

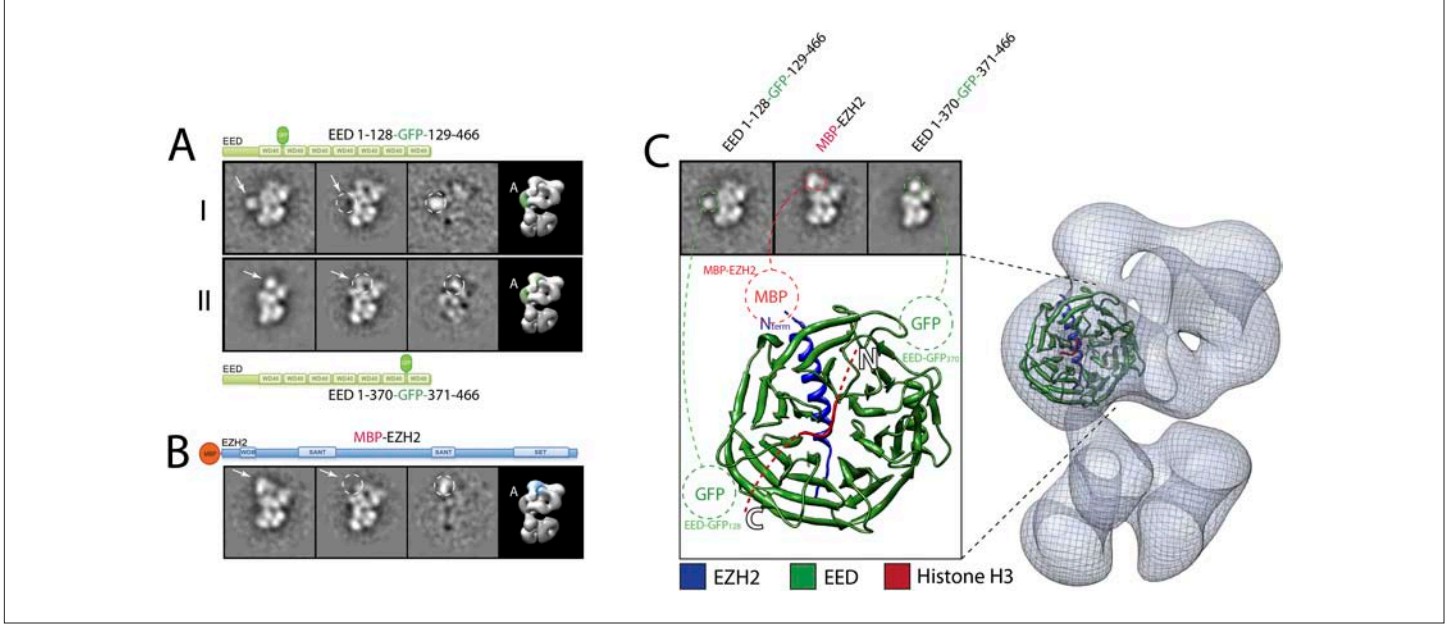

**Figure 7**. Localization and arrangement of the subunit EED within the PRC2 electron density map: Three different internal GFP tags (EED-EGFP128, EED-EGFP370 and MBP-Ezh2) were used to determine the correct position of the EED-Ezh2 crystal structure (PDB: 2QXV) within the EM reconstruction. (**A and B**) For each study, a cartoon indicates the position of the tag within the protein. MBP tags are indicated in red, GFP tags in green. The far left panel represents the reference-free 2D class obtained from the labeled sample. The middle left panel shows the corresponding class for the unlabeled sample. The middle right panel was calculated by subtracting the labeled reference-free class from the unlabeled average (labeled and unlabeled). Only differences with a standard deviation greater than 3, are considered significant. The far right panel includes a representative 3D view of the complex, the localized density color-coded and the assigned Lobe indicated. (**C**) Top panel: 2D class averages for each mutant indicates the position of the specific tag on the molecule. Bottom panel: fitting of the X-ray structure within the EM density based on the specific position of the tags. Docking of the Histone H3 based on its crystal structure in complex with the subunit EED (PDB: 3IIW) is also indicated.

of this construct showed additional density corresponding to an MBP tag protruding from Lobe A (*Figure 7B*). Based on the positions of the two internal GFP tags for EED and the MBP tag for Ezh2, we were able to accurately orient the EED-Ezh2 crystal structure within the 3D EM density. This information, together with the crystal structure of histone H3-bound EED (*Margueron et al., 2009*), suggests that the canonical histone H3-binding site is located in a cavity at the intersection of Lobe A and Arm 2 (*Figure 7C*).

With the EED's WD40 domain assigned to Lobe A, we predicted that Lobe D would correspond to the WD40 propeller of RbAp48. To test this idea, we reconstituted complexes carrying an N-terminal MBP or and internal or C-terminal GFP tag on RbAp48, but none of these fusion constructs produced a stable and stoichiometric complex. To overcome this limitation, we again took advantage on the known interaction between Nurf55 (*Drosophila* ortholog of RbAp48) and Suz12 that has been structurally characterized (*Nowak et al., 2011*; *Schmitges et al., 2011*). We generated a complex bearing an internal GFP tag immediately after the alpha helix of Suz12 that interacts with RbAp48 (Suz12-GFP$_{123}$). In this particular case, the canonical view was not as well represented and the label was more clearly seen in a different orientation (*Figure 8A*). The GFP can be seen localizing to Lobe D, confirming that Suz12 interacts with the RbAp48 propeller at this position.

Using the local rigid-body fitting algorithm implemented in the UCSF Chimera software package (*Pettersen et al., 2004*), we docked the crystal structures of the *Drosophila* RbAp48-Suz12 complex (PDB: 2YB8) into our density map (*Figure 8B*). The WD40 propeller and protruding N-terminal alpha-helix of RbAp48 were accommodated within this region with high fidelity. The docked crystal structure of the histone H3-bound RbAp48 (*Nowak et al., 2011*; *Schmitges et al., 2011*) provides further insight into histone binding. Attached to RbAp48 at Lobe D, the histone would face Lobe C (*Figure 8B*), suggesting that the histone H3 tail is situated in a head-to-tail orientation, running vertically from the bottom end of the complex in Lobe D towards Lobe B.

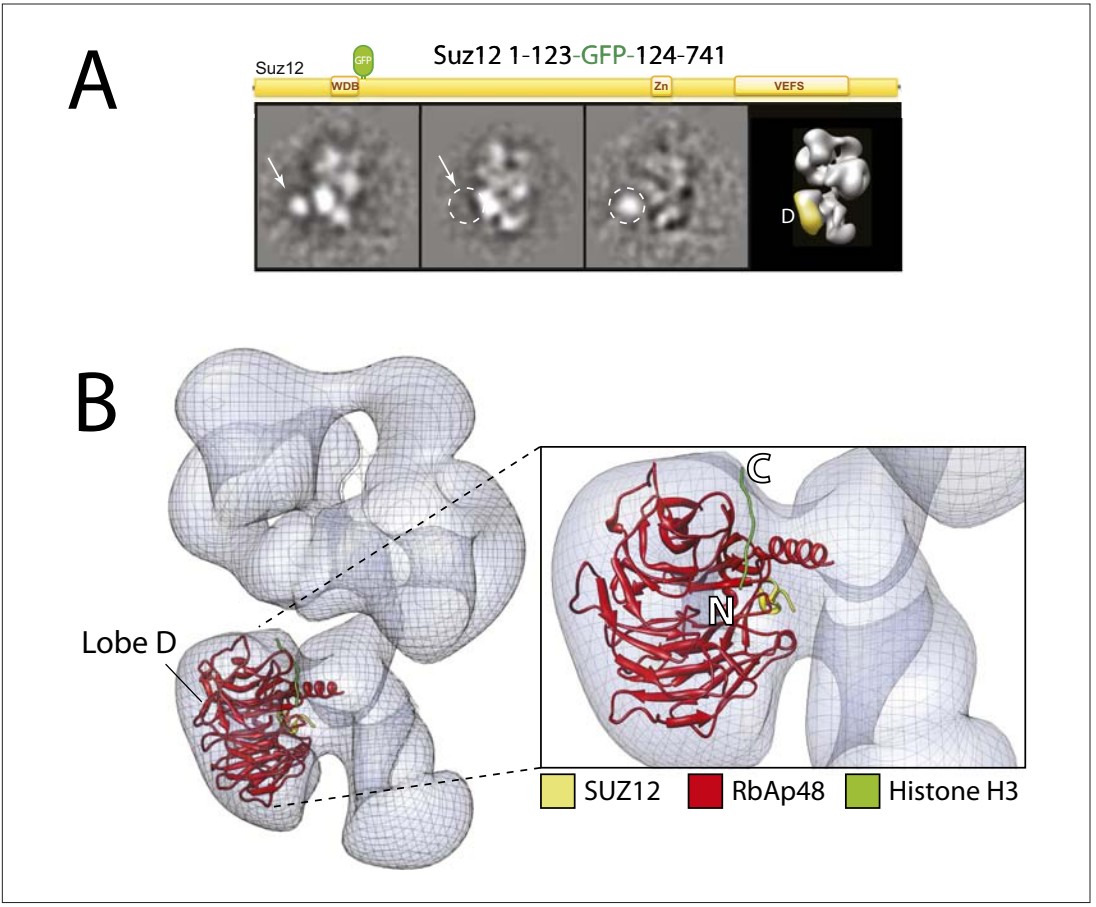

**Figure 8**. Localization of the subunit RbAp48 within the PRC2-AEBP2 electron density map. (**A**) Reference-free 2D class of the labeled, unlabeled sample, difference map and 3D view of the complex with assigned localization. (**B**) Fitting of the Nurf55-Suz12 crystal structure within the density of Lobe D. The density has been automatically placed using the local localization algorithm implemented in the UCSF Chimera software (**Pettersen et al., 2004**). Docking of the Histone H3 based on its crystal structure in complex with the RbAp48 homologue subunit Nurf55 (PDB: 2YBA) is also indicated.

Next, and in order to locate within PRC2 the different domains of the catalytic Ezh2 subunit, we again used an internal GFP labeling strategy in addition to our MBP N-terminal label (**Figure 9A**, panel I). To localize the two SANT domains, we reconstituted PRC2-AEBP2 complexes containing either a GFP label downstream of the first SANT domain (Ezh2-GFP$_{303}$, **Figure 9A**, panel II; see also **Figure 9—figure supplement 1**) or immediately upstream of the second SANT domain (Ezh2-GFP$_{436}$, **Figure 9A**, panel III). The location of the GFP extra density suggests that the first SANT domain is positioned at the intersection of Lobe A and Arm 1, and the second at the intersection of Arm 1 and Lobe B. The proximity of these two SANT domains is in very good agreement with our cross-linking analysis (**Figure 5**).

The methyltransferase SET domain of Ezh2 was located by reconstituting two complexes, one bearing an internal GFP tag immediately upstream of this domain (Ezh2-GFP$_{595}$, **Figure 9A**, panel IV; see also **Figure 9—figure supplement 1**, **Movie 1**) and another with a C-terminal GFP tag (Ezh2-GFP, **Figure 9A**, panel V). Analysis of the 2D class averages indicates that the SET domain begins at Lobe B and terminates above EED in Lobe A. **Figure 9B** summarized our Ezh2 labeling results (proposed chain path in inset): the two SANT domains span Arm 1 and Lobe B, while the C-terminal SET domain is located between Lobes A and B. Our identification of a cross-link between the first Ezh2 SANT domain and the N-terminus of EED (**Figure 5**) further suggest that the N-terminal region of EED localizes to Arm1 (**Figure 11**). In agreement with our tagged-based localization, crystal structures for homologous SANT and domains and SET domains (PDBs: 3HM5 and 3H6L) fit well within the assigned lobes in the EM structure (**Figure 11** and **Movie 2**).

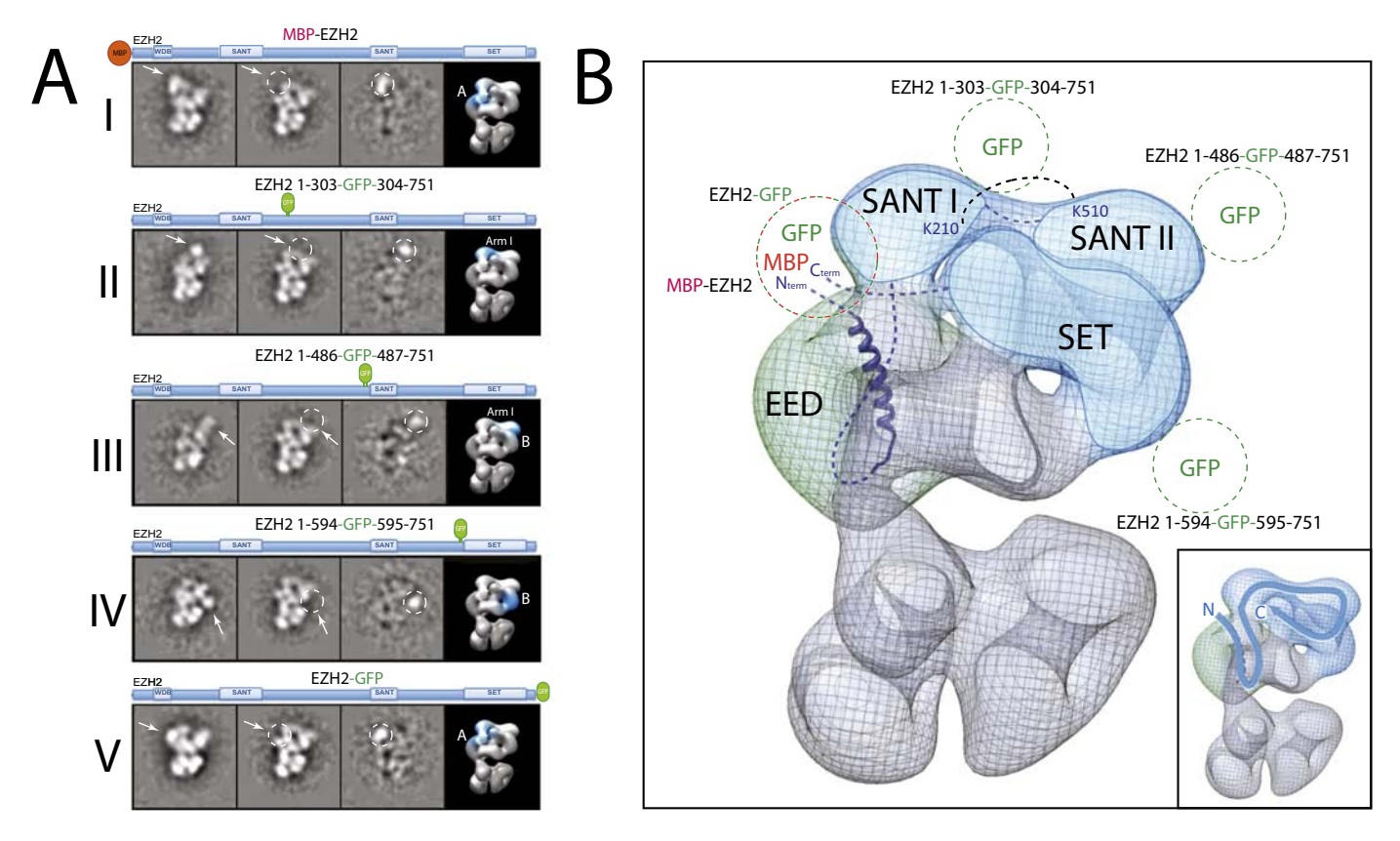

**Figure 9**. Localization and arrangement of the subunit EZH2 within the PRC2-AEBP2 electron density map. (**A**) Reference-free 2D class of the labeled, unlabeled sample, difference map and 3D view of the complex with assigned localization for different MBP/GFP fusion mutants of EZH2. (**B**) Assignment of EZH2 subunit domains to electron density map based on the specific position of the tags. The crosslink between K210 and K510 is shown. The EZH2 protein chain path within the PRC2-AEBP2 complex is indicated in the inset (See also **Figure 9—figure supplement 1**).

The following figure supplements are available for figure 9:

**Figure supplement 1**. Subclassification of particles from labeled complexes illustrate the flexibility of the tag.

In order to localize Suz12 we generated complexes containing tags at either the N- or C-termini of this subunit. However, neither of them resulted in clearly identifiable densities by 2D analysis, probably because these two regions are predicted to be unstructured and the tags were too mobile. While mutant complexes carrying internal GFP tags at the Zn finger and VEFS domains resulted in unstable complexes, two others were amenable for structural studies. The first was a mutant complex containing a GFP tag in the central portion of Suz12 (Suz12-GFP$_{269}$) that exhibited additional density near Lobe C (**Figure 10A**, panel I). The second complex contained a C-terminal truncation of Suz12 after the VEFS domain, with a GFP tag at the truncated terminus (Suz12-ΔC-GFP). This complex showed extra density extending from Lobe B (**Figure 10A**, panel II). Together, our labeling data suggest that Suz12's N-terminal region is located at Lobe C, and terminates at Lobe B near the SET domain of Ezh2, most likely passing through Arm2 (**Figure 10B**).

Finally, to determine the spatial organization of the cofactor AEBP2 as it binds to PRC2, we reconstituted different complexes carrying protein tags in AEBP2 at the N- and C-termini (MBP-AEBP2; AEBP2-EGFP) and in the two loops connecting the Zn finger domains (AEBP2-GFP$_{84}$ and AEBP2-GFP$_{115}$). With the exception of MBP-AEBP2, which failed to form a stable complex, all the resulting complexes were suitable for EM analysis. AEBP2-GFP$_{84}$ (**Figure 10A**, panel III) and AEBP2-GFP$_{115}$ (**Figure 10A**, panel IV) both localized near Lobe A and Arm 1. This location, in the vicinity of EED and the first SANT domain of Ezh2 (**Figure 10B**), agrees with our cross-linking data (**Figures 5 and 10C**). 2D analysis of the

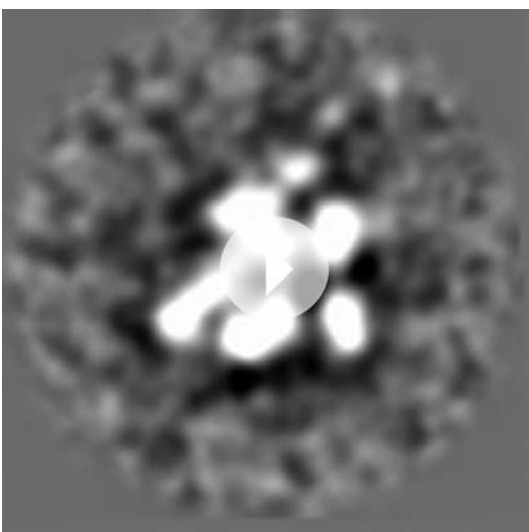

**Movie 1**. Movement of GFP tag.

AEBP2-EGFP–containing complex shows that the C-terminal region of AEBP2 is located between Lobes A and D, possibly within Arm 2 (*Figure 10A*, panel V). Importantly, our cross-linking data also indicate that the AEBP2 C-terminus makes extensive interactions with the central portion of Suz12 (304–416), suggesting that these two segments might co-localize to Arm 2 and into Lobe D (*Figures 5 and 10C*). Thus, AEBP2 runs in the opposite direction of Suz12, with its central portion located near Arm 1 and Lobe A, from where it continues along the PRC2 structure, terminating near Arm 2 and the top of Lobe D (*Figure 10B*). The inset in *Figure 10B* summarizes the proposed chain paths for AEBP2, and SUZ12.

Our labeling studies indicate that the PRC2-AEBP2 complex contains a single copy of each subunit, as we observed only one additional density within each of the individually tagged complexes. These data also confirm that the determined structure contains all the purified components. The proposed overall architecture for the PRC2-AEBP2 complex is summarized in *Figure 11*. The available crystal structures for EED and RbAp48 WD40 domains, as well as those for homologues of the SANT, SET and Zn finger domains are included.

In conclusion, using a combination of redundant and consistent data from protein biochemistry, chemical cross-linking and mass spectrometry, electron microscopy, and crystal structure docking, our study provides a robust model of the overall architecture and domain organization of the PRC2-AEBP2 complex. The unprecedented detail in our architectural map of the complex should serve as an invaluable structural framework for past and future functional studies of this essential silencing complex.

## Discussion

In higher eukaryotes, the Trithorax (TrxG) and Polycomb groups (PcG) of protein complexes play a fundamental role in orchestrating the transcriptional activation or silencing of the genome. These genomic 'on' and 'off' switches are primarily controlled through the trimethylation of H3K4 (H3K4-me3), H3K36 (H3K36-me2/3), and H3K27 (H3K27-me3). The PRC2 complex is an important member of the PcG family, and a large number of biochemical studies have suggested models for its role in mediating gene repression (reviewed in *Margueron and Reinberg, 2011*). Characterizing the spatial arrangement of the PRC2 components is a fundamental step towards understanding the molecular mechanism underlying PRC2's regulated histone methylation and its role in gene silencing. However, structural studies so far have been limited to two isolated domains out of the four PRC2 protein subunits, severely restricting our understanding of the molecular mechanism of this fundamental complex. Here, by using EM in conjunction with chemical cross-linking and mass spectrometry, we provide the complete PRC2 architecture, describing the structure of this complex bound to its co-factor AEBP2. We present a detailed map of localizations and interactions of its constitutive subunits, integrating previous structural and biochemical studies into a more comprehensive molecular model of PRC2 activity.

Previous reports have shown that the PRC2 subunit EED is able to bind the chromatin repressive mark H3K27me3 in trans, via its WD40 domain, thus enhancing PRC2 mediated methylation of H3K27 in oligonucleosomes (*Margueron et al., 2009*; *Xu et al., 2010*). This activation results in propagation of the repressive chromatin state to neighboring nucleosomes containing newly incorporated unmodified histone H3 (*Hansen et al., 2008*; *Margueron et al., 2009*). On the other hand, the studies of Schmitges and colleagues suggested that binding of the chromatin activation marks H3K4me3 and H3K36me2-3, inhibits PRC2 activity if placed in cis on the same tail containing the target lysine H3K27 (*Schmitges et al., 2011*). Additionally, it has been proposed that the subunit Suz12, and in particular its VEFS domain, is responsible in mediating this inhibition (*Schmitges et al., 2011*). Since this inhibitory

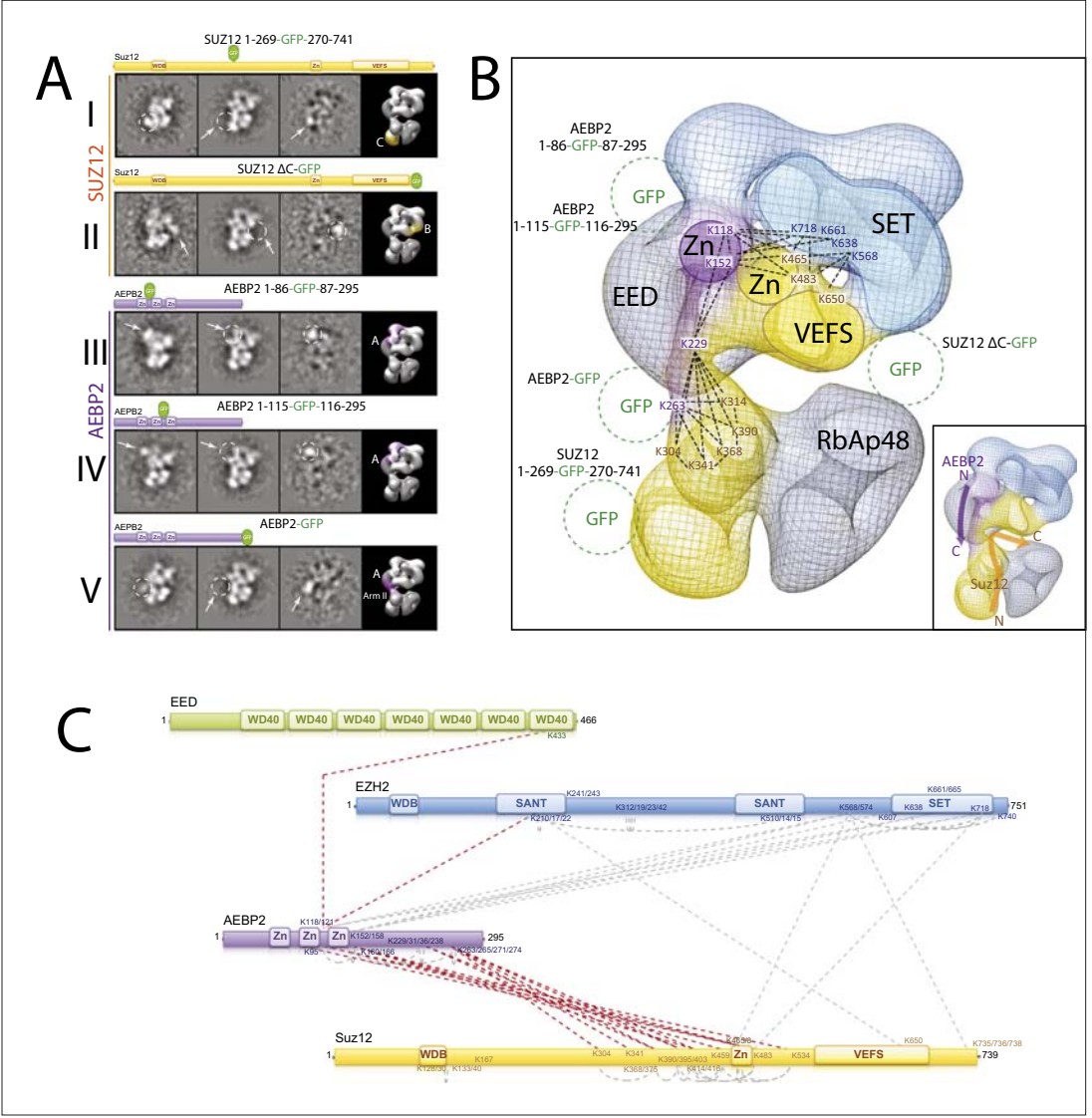

**Figure 10**. Localization and Arrangement of the subunits Suz12 and AEBP2 in the PRC2-AEBP2 electron-density map. (**A**) Reference-free 2D class of the labeled, unlable sample, difference map and 3D view of the complex with assigned localization for different GFP fusion mutants of Suz12 and AEBP2. (**B**) Assignment of Suz12 and AEBP2 subunit domains to electron density map based on the specific position of the tags. Suz12 and AEBP2 protein chains paths within the PRC2-AEBP2 complex are indicated in the inset. Aminoacidic numbers indicate cross-linked residues in the contest of the electron density map. (**C**) Isotopic cross-linking of PRC2 complex. Connections in red indicate the cross-linked aminoacids referred in the text.

effect can be reversed by the addition of H3K27me3 containing peptides, it has been thought that PRC2 can simultaneously integrate inhibitory and activating chromatin marks tuning its enzymatic activity based on the surrounding chromatin state (*Schmitges et al., 2011*).

The results we present here make sense of these previous biochemical findings within the context of the functionally identified architecture of the PRC2 complex, allowing us to propose models that serve as hypotheses for further testing. Our structure shows that the Ezh2's methyltransferase domain forms a structural core with the two activity-controlling elements of PRC2, the WD40 domain of EED and the VEFS domain of Suz12 (*Margueron et al., 2009*; *Schmitges et al., 2011*). This suggests that the proposed regulatory mechanisms of these domains on PRC2 activity could be structurally supported by their physical interaction with the SET domain.

SANT domains have been shown to be important in coupling histone-tail binding to enzymatic activity in many chromatin regulating complexes (*Boyer et al., 2004*). Additionally, it is known that

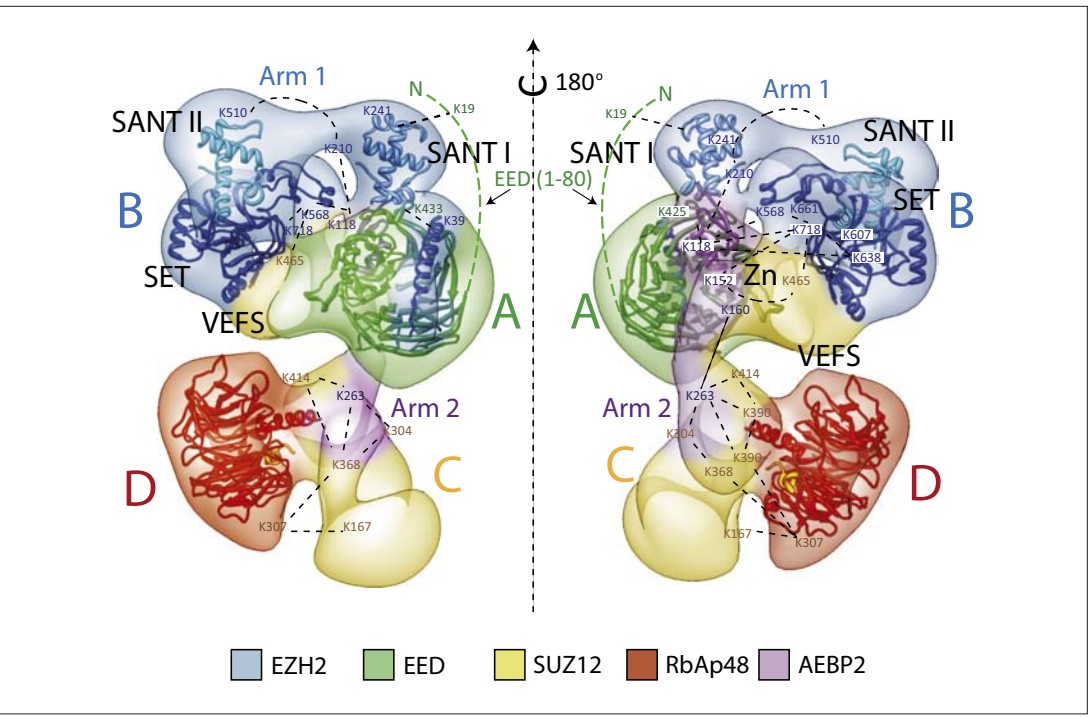

**Figure 11**. Overall Architecture of the PRC2 Complex. Model of the Prc2 complex from a combination of biochemical data, mass spectrometry, and structural biology data. Docking of crystal structure for EED and RbAp48 WD40s (PDB: 2QXV; 2YB8) are indicated respectively in green and red. Docking of crystal structures of homologue SANT, SET and Zn finger domains (PDB: 3HM5, 3H6L and 2VY5) are shown in blue and purple. Approximate positions of crosslinking sites are also indicated. See also **Movie 2**.

histone H3 binds to the N-terminal tail of EED (*Tie et al., 2007*). Our structure and cross-linking studies show an intricate network of interaction between the N-terminal and WD40 regions of EED and SET and SANT domains of Ezh2. These SANT domains may couple EED-mediated binding of methylated histone H3 to the methyltransferase activity of Ezh2 (*Figure 12A*), suggesting that complex activity could be influenced through incorporation of EED isoforms that differ in their N-terminal regions.

At the opposite end of the core module, the VEFS domain of Suz12 and the SANT and SET domains of Ezh2 together interact with two Zn fingers from AEBP2 and Suz12 (*Figure 12B*). The position of the Suz12 VEFS domain with respect to the Ezh2 SET domain suggests how the H3K4me3 and H3K36me2/3 inhibition via Suz12 could be transferred to Ezh2, regulating its methyltransferase activity. The close proximity of these elements could also explain why the gene activation marks need to be on the tail being modified in order to block methyltransferase activity.

In summary, our structure offers a three dimensional framework showing how various elements of the PRC2 complex involved in regulated histone-binding can act either independently or synergistically in response to different chromatin environments. Importantly, our PRC2 subunit organization is consistent with previously proposed models of PRC2 activity (*Margueron et al., 2009*; *Schmitges et al., 2011*): when chromatin domains are targeted for repression, PRC2 is recruited to deposit the repressive H3K27me3 histone mark at nucleosome-associated regions of DNA. Existing H3K27me3 marks are recognized by EED and, possibly in cooperation with Ezh2's SANT domains, Ezh2 methyltransferase activity is increased (*Figure 12A*). To ensure that H3K27 trimethylation is limited only to the repressed target, PRC2 is capable of repressing its own enzymatic activity. When PRC2 comes into contact with the boundary of an active gene, the interaction between H3K4me3, H3K36me2, and H3K36me3 with the VEFS domain of Suz12, could be propagated to the Ezh2 subunit, blocking PRC2 activity (*Figure 12B*).

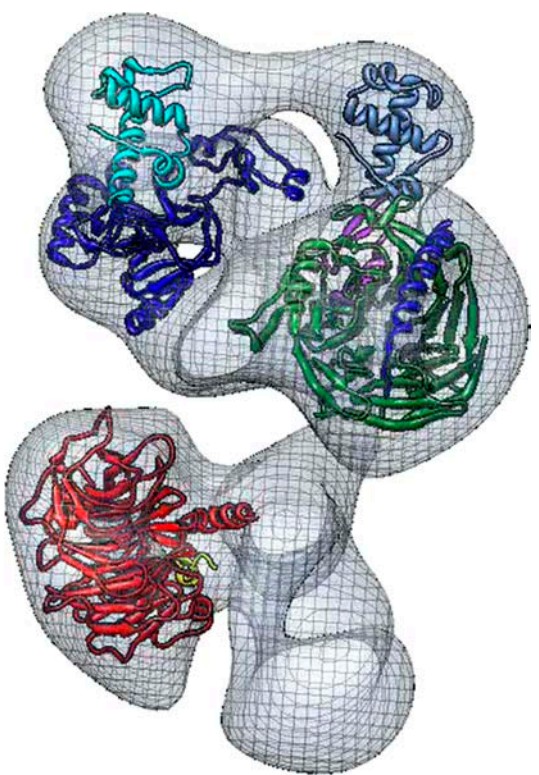

**Movie 2**. 3D reconstruction of the human PRC2-AEBP2 complex.
The following source code are available for movie 2:
**Source code 1**: Overall architecture of the PRC2 complex.

Previous studies have shown that PRC2 favors di- and oligonucleosome substrates over mononucleosomes, octamers, or histone H3 peptides (*Cao and Zhang, 2004a*; *Kuzmichev et al., 2004*; *Martin et al., 2006*). Molecular explanations for this substrate preference have been largely hypothetical in the absence of any structural information. The positioning of the different subunits within the PRC2 structure suggests a model of how PRC2 could interact with a dinucleosome, by placing the regions interacting with histone tails in opposite sides of the complex, thus allowing interaction with two nucleosomes simultaneously, without any steric hindrance. *Figure 13* suggests a possible arrangement illustrating this point that also agrees with the proposed binding of AEBP2 to nucleosomal DNA (*Kim et al., 2009*). In such arrangement, EED binding to one nucleosome would position the histone H3 tail from the second nucleosome in close proximity to the Ezh2 SET domain (*Figure 13*).

In conclusion, the human PRC2 structure presented here provides us with the first full picture of the molecular organization of this fundamental complex and offers an invaluable structural context to understand previous biochemical data. Furthermore, the functional mapping of different activities within the physical shape of the complex leads to novel, testable hypotheses on how PRC2 interacts with chromatin that should inspire future research of PRC2 function and regulation. Given the similarity in sequence between PRC2 components from different species, we expect the molecular architecture that we described for human PRC2 to be conserved throughout higher eukaryotes.

# Materials and methods

## PRC2 complex reconstitution and purification

Sequences corresponding to the full-length EED, RbAp48, Ezh2, Suz12 and AEBP2 (see *Figure 1A* for domain schematics for these proteins) were amplified from human full-length cDNA clones. AEBP2 was subcloned in frame with a N-terminal StrepII tag into the pFastBac1 shuttle vector. All the other components were subcloned in the same vector in frame with a N-terminal hexahistidine tag. For PRC2 component localization, all the subunits were sub-cloned into a modified pFastBac1 vector in frame with either an N-terminal maltose binding protein or an internal or C-terminal GFP. All clones were verified by DNA sequencing. Recombinant baculoviruses were generated in Sf9 cells using the Bac-to-Bac kit (Invitrogen, Grand Island, NY, USA) according to manufacturer's recommendations. For protein expression, High5 cells were grown in suspension culture in ESF921 serum-free medium (Expression Systems, Davis, CA, USA) and infected with the right combination of viruses at a density of $1.0 \times 10^6$ ml$^{-1}$. Cells were harvested 60–72 hr post-infection and lysed by sonication in 25 mM Hepes pH 7.5, 138 mM NaCl, 10% Glycerol, 0.05% Nonidet-P40 (NP40) and 1 mM Dithiothreitol (DTT) (supplemented with protease inhibitors). All samples were affinity-purified on Strep-Tactin Superflow Plus resin (Qiagen, Valencia, CA, USA) using an N-terminal Strep-tag II tag on the AEBP2 subunit. The complex was eluted from the resin by competition with lysis buffer containing 10 mM Desthiobiotin. The complex was then immediately subjected to size exclusion chromatography (SEC) on a calibrated Superose 6 PC 3.2/30 column (GE Healthcare, Uppsala, Sweden) at 4°C equilibrated in SEC buffer (25 mM Hepes pH 7.5,

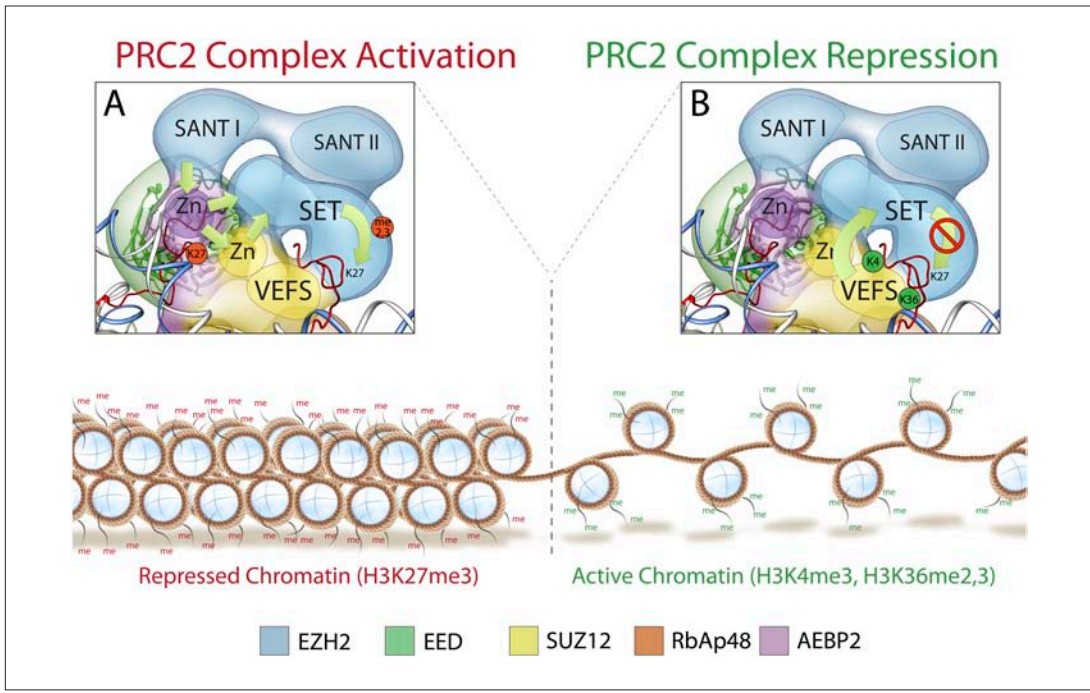

**Figure 12**. Mechanism and allosteric regulation of PRC2 during gene silencing. (**A**) At loci of compact and repressed chromatin, H3K27-me3 marks are recognized by EED. This binding is signaled via the SANT domains to the SET domain increasing the methyl-transferase activity of Ezh2, strengthening the chromatin compaction. (**B**) At loci of open and actively transcribed chromatin, H3K4me3 and H3K36me2,3 are recognized by the VEFS domain of Suz12 and transferred to Ezh2, with an allosteric regulation that blocks Ezh2's enzymatic activity.

138 mM NaCl, 10% Glycerol, 0.05% NP40, 1 mM Tris(2-carboxyethyl) phosphine hydrochloride [TCEP]). To preserve protein stability for structure determination, Prc2 protein complexes were cross-linked with 0.015% Glutaraldehyde for 10 minutes and the reaction stopped with 1M Tris, pH 8.0. Protein elution was monitored at 280 nm and protein fractions were collected and analyzed by SDS-PAGE and Coomassie brilliant blue staining.

## Chemical cross-linking using isotope labeled cross-linkers and mass spectrometric identification of cross-linked peptides

Human PRC2 Complex (10 µg of protein equivalent to 36 pmol) was mixed with a 600X excess of isotope-labeled cross-linker di-(sulfosuccinimidyl)-glutarate (1:1 mixture of light DSSG-D0 and heavy DSSG-D6) (Creative Molecules, Andover, MA, USA) in a final volume of 50 µl of 10 mM Hepes, pH 7.5, 138 mM NaCl at room temperature. The reaction was stopped after 30 min by adding 5 µl of 1 M ammonium bicarbonate.

Cross-linked proteins were reduced with 5 mM TCEP (tris (2-carboxyethyl) phosphine; Thermo Scientific, Rockford, IL, USA) at 37°C for 15 min and alkylated with 10 mM iodoacetamide (Sigma-Aldrich, St. Louis, MO, USA) for 30 min in the dark. Proteins were digested with trypsin (Promega, Madison, WI, USA) at an enzyme to substrate ratio of 1:50 (wt/wt) at 37°C for 2 hr and, after a second addition of trypsin (1:50 wt/wt), over night. Peptides were acidified with 1% trifluoroacetic acid (TFA; Sigma-Aldrich, St. Louis, MO, USA) and purified by solid-phase extraction (SPE) using C18 cartridges (Sep-Pak; Waters, Milford, MA, USA). The SPE eluate was evaporated to dryness and reconstituted in 20 µl of SEC mobile phase (water/acetonitrile/TFA, 70:30:0.1). 15 µl were injected on a GE Healthcare (Uppsala, Sweden) Äkta micro system. Peptides were separated on a Superdex Peptide PC 3.2/30 column (300 × 3.2 mm) at a flow rate of 50 µl min⁻¹ using the SEC mobile phase. Two-minute fractions (100 µl) were collected into 96-well plates.

LC-MS/MS analysis was carried out on an Eksigent 1D-NanoLC-Ultra system connected to a Thermo LTQ Orbitrap XL mass spectrometer equipped with a standard nanoelectrospray source. SEC fractions

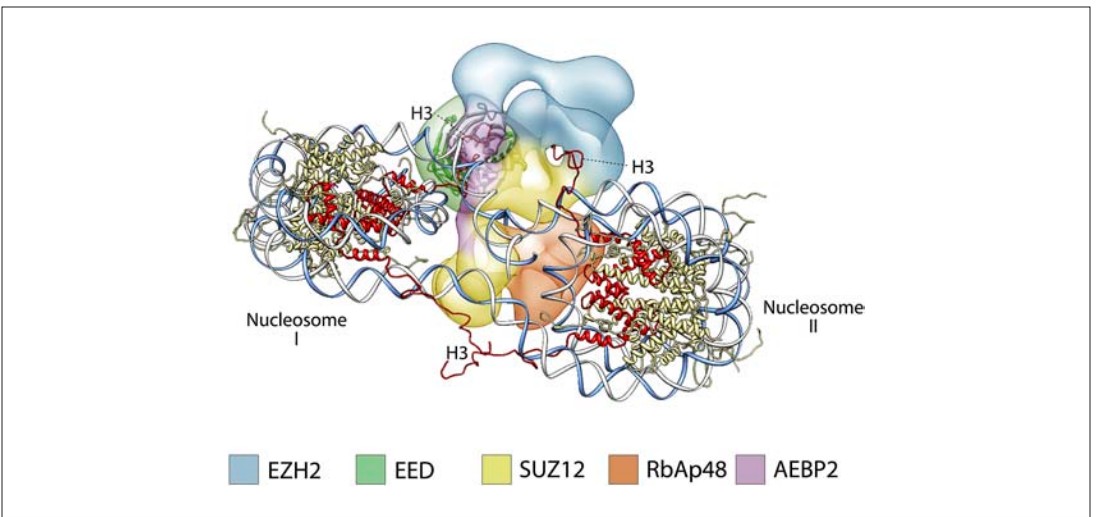

**Figure 13**. A proposed possible model for the binding of the PRC2-AEBP2 complex to a di-nucleosome.

were reconstituted in mobile phase A (water/acetonitrile/formic acid, 97:3:0.1). A fraction corresponding to an estimated 1 µg was injected onto a 11 cm × 0.075 mm I.D. column packed in house with Michrom Magic C18 material (3 µm particle size, 200 Å pore size). Peptides were separated at a flow rate of 300 nl min$^{-1}$ ramping a gradient from 5% to 35% mobile phase B (water/acetonitrile/formic acid, 3:97:0.1).

Cross-linked peptides were identified using an in-house version of the dedicated search engine, xQuest (*Rinner et al., 2008*). Tandem mass spectra of precursors differing in their mass by 6.037660475 Da (difference between DSSG-d0 and DSS-d6) were paired if they had a charge state of 3+ to 8+ and were triggered within 2.5 min of each other. These spectra were then searched against a pre-processed *.fasta* database. A valid identification of the cross-linked peptides required at least four bond cleavages in total or three in a series for each peptide and a minimum peptide length of six amino acids.

## Electron microscopy sample preparation and data collection

A 4 µl aliquot of purified sample was immediately placed onto a continuous carbon grid that had been plasma cleaned in a 75% Ar/25% $O_2$ atmosphere for 20 s in a Solarus plasma cleaner (Gatan, Inc, Pleasanton, CA, USA). After 30 s of incubation on the grid at room temperature, the sample was negatively-stained with a solution of 2% uranyl formate and blotted dry. Samples were imaged using a Tecnai F20 Twin-transmission electron microscope operating at 120 keV at a nominal magnification of 80,000× (1.52 Å/pixel at the detector level) using a defocus range of −0.6 to −1.3 µm. Images were automatically recorded with an electron dose of 20e-/Å$^2$ using the Leginon data collection software (*Suloway et al., 2005*) on a Gatan 4096 × 4096 pixel CCD camera (15 µm pixel size). For the random conical tilt (RCT) dataset, images were collected at −60° and 0°.

## Image processing

All image pre-processing and two-dimensional classification was performed in the Appion image-processing environment (*Lander et al., 2009*). A general scheme was utilized for image analysis to deal with the large number of datasets acquired, as well as to minimize user bias. Particles were initially selected from the 0° tilt micrographs of the PRC2-AEBP2 complex using a difference of Gaussians (DoG) transform-based automated picker (*Voss et al., 2009*), and extracted using a 224 × 224-pixel box size. Iterative multivariate statistical analysis (MSA) and multi-reference alignment (MRA) of the extracted particles provided representative 2D views of the PRC2 complex, which served as templates for all subsequent automated particle selection.

The generalized processing scheme utilized for processing was as follows. The ACE2 and CTFFind programs (*Mindell and Grigorieff, 2003*; *Mallick et al., 2005*) ran concurrently with data collection to estimate the contrast transfer function (CTF) of the micrographs, as well as to provide a quantitative measurement of the quality of the imaging. At the same time, a template-based particle picker using the

representative views of PRC2 (described above) located particles on the micrograph. At the end of data collection, the phases of the micrograph were corrected with ACE2, and individual particles were extracted using a box size of 224 × 224 pixels and decimated by a factor of 2. Any pixels whose values were above or below 4.5 sigma of the mean pixel value were removed using the XMIPP normalization function (*Scheres et al., 2008*). In order to remove inappropriately selected protein aggregates, stain crystals, carbon edges, or other forms of contamination, particles whose mean or standard deviation deviated significantly from the norm were removed. The remaining particles were subjected to a single round of MSA and classification (*Ogura et al., 2003*), in which thousands of class averages were generated with ~20 particles per class. The resulting class averages were manually inspected to remove remaining aggregation, contamination, false positive particle selections, and very small particles potentially corresponding to incomplete PRC2 complexes. Following these particle-cleaning steps, the remaining 39,527 particles were subjected to several rounds of MSA and MRA using the IMAGIC software package (*van Heel et al., 1996*) to produce detailed views of the PRC2 complex with high signal-to-noise ratios.

## Analysis of tagged complexes

For subunit localization, reference-free 2D class averages from each tagged dataset were generated. We concentrated any further efforts on classes corresponding to the canonical view. The full set of tagged particles corresponding to this projection view gives rise to a class average with very good signal for the complex, as well as some extra density, corresponding to the tag, which is distinctive but typically smeared to different extents for different samples due to the flexibility of the tag. For a clearer visualization of the tag, we subclassified particles within this view. Two examples of the subclassification are shown for GFP-tagged complexes in *Figure 9—figure supplement 1*. The pivoting of the tag density can be clearly seen in these classes and in *Movie 1*. The percentage of subclasses clearly showing the tag ranges from 100% for a significant number of labels to 57% for the least favorable case (likely due to proteolysis). For the generation of the figures in the main text of the paper we chose to show a single subclass average with a larger number of particles that accurately resembles the full ensemble, but that more clearly shows the density of the tag, nicely corresponding to the mass expected for proteins of their size. These classes were then compared to the wild-type PRC2 class averages through cross-correlation using the SPIDER 'AP SH' command. After alignment, the matching class averages for the untagged and tagged complexes were individually normalized and then the PRC2 class averages were subtracted from the PRC2-tagged class averages. The densities seen in the difference maps were at least 3-standard deviations above the mean, showing the location of tagged-PRC2 subunits.

## Ab initio reconstruction using random conical tilt and 3D reconstruction refinement

In order to generate an initial three dimensional model of the negatively-stained PRC2 complex, 190 tilt-pair images (60° and 0°) were collected using the same imaging conditions as described in the previous section. These data were collected in an automated fashion using the RCT-Raster application in Leginon (*Yoshioka et al., 2007*). Particles were automatically selected using the DoG particle picker (*Voss et al., 2009*), and the tilt axis, as well as the particle-pair assignments, were calculated automatically with the TiltPicker module of Appion TiltPicker (*Voss et al., 2009*). Particles were extracted in the same manner as described in the previous section, resulting in 6075 particle pairs. The XMIPP ML2D program was used to generate 20 reference-free 2D averages from the 0° micrographs, each class containing between 150 and 800 particles (*Scheres et al., 2008*). Generation of more than 20 classes did not reveal conformers that were notably different from the reconstructions resulting from the initial classes, and suffered in resolution due to the decreased number of particles attributed to each class. SPIDER routines integrated into Appion were used to generate 3D RCT reconstructions for each of these class averages.

In order to minimize missing-cone artifacts inherent to the RCT methodology from influencing subsequent projection-matching-based reconstructions, and assess the homogeneity of the PRC2 conformation represented in the class averages, all the RCT reconstructions were used as references to assign Euler angles to 1000 high signal-to-noise reference-free class averages from the 39,527-particle untilted dataset using projection-matching, in lieu of merging the RCT reconstructions into a single density. The class averages were then back-projected according to their assigned Euler angles to provide a three-dimensional model based solely on 0° particles data. The resulting 3D reconstructions, albeit at low resolution, were highly consistent, indicating that there is one major architectural state of the complex. We used the reconstruction obtained from the more populated RCT class as the initial

model for refinement against single particles of the full 0° data set. Since potential end-on views of the complex were not considered during structure determination, as they could not be distinguished from partial complexes, we relied on 'tomographic' coverage of the views along the long axis of the structure. This refinement was performed using an iterative projection-matching script that makes use of libraries from the SPARX and EMAN2 image processing packages (*Baldwin and Penczek, 2007*; *Tang et al., 2007*).

## Acknowledgements

We are grateful to S. Sun, M Jinek, J. Querol Audi, M Cianfrocco, T. Houweling, A. Costa, E. Arias Palomo and Ann Fisher (UC Berkeley Tissue Culture Facility) for technical support. We are thankful to Denny Reinberg, Andrea Carfi and Diego Pasini for critical reading of the manuscript.

## Additional information

### Funding

| Funder | Grant reference number | Author |
| --- | --- | --- |
| NIGMS | GM63072 | Eva Nogales |
| European Union Seventh Framework Program PROSPECTS | HEALTH-F4-2008-201648 | Franz Herzog |
| ERC advanced grant "Proteomics v3.0" | 233226 | Ruedi Aebersold |
| Damon Runyon Cancer Research Foundation Fellowship | | Gabriel C Lander |
| Marie Curie Fellowship | | Franz Herzog |
| EMBO Fellowship | | Alessio Maiolica |
| Howard Hughes Medical Institute | | Eva Nogales |

The funders had no role in study design, data collection and interpretation, or the decision to submit the work for publication.

### Author contributions

CC, Conception and design, Acquisition of data, Analysis and interpretation of data, Drafting or revising the article; GL, Acquisition of data, Analysis and interpretation of data, Drafting or revising the article; AM, Conception and design, Acquisition of data, Analysis and interpretation of data, Drafting or revising the article; FH, Acquisition of data, Analysis and interpretation of data, Drafting or revising the article; RA, Conception and design, Analysis and interpretation of data, Drafting or revising the article; EN, Conception and design, Analysis and interpretation of data, Drafting or revising the article.

## Additional files

### Supplementary files
• Supplementary file 1. Inter- and intramolecular crosslinking.

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
