## [Author Response]

•*The original images show a heterogeneous population of particles and in particular smaller particles that suggest that some particles may dissociate. We also notice that little information is given regarding the stoichiometry of the complex. However the data is consistent and convincing. The authors should describe whether all particles were considered or whether some were rejected. If the latter was the case, the rejection criteria should be communicated*.

We thank the reviewer for pointing out the omission of this section of the text. The manuscript now includes this information in the results section. We mentioned that, as we affinity purified the PRC2-ABP2 complex using a Strep tag onto AEBP2, we cannot exclude the presence of a small excess of this subunit in our preparation. Regarding the EM images, we noticed the presence of smaller particles. We interpreted them as either end-on views or partial complexes, most likely the AEBP2 itself, being used for tag-based purification. Those particles were consequently not considered during the 3D data processing, and we relied on a good “tomographic” coverage of the views around the long axis of the structure. It should also be noted that the labeling studies clearly indicate that the structure described contains all the subunits.

•*The conclusions of the manuscript, and in particular the functional consequences for the interaction of PCR2 with nucleosomes, are highly speculative. The histone tails that interact with the WD40 repeats of PCR2 are extremely flexible and one cannot conclude at this stage that PRC2 will bind between two adjacent nucleosomes. A part of the discussion concerns the interaction with the so-called linker histone H1. H1 interacts mainly through its globular domain with the dyad axis of the core nucleosome and has little interaction, except for its N-and C termini with the linker DNA. Once bound to nucleosomes, H1 induces an important condensation of the chromatin and it is unlikely that PCR2 can still bind between two successive nucleosomes. Therefore this part of the discussion should be removed*.

We thank the reviewer for pointing out the issues in this part of the discussion. We have now completely removed the portion regarding the Histone H1 binding and rephrased, in a more cautious and succinct way, the section concerning the nucleosome binding. We still indicate that our structure does suggest a possible arrangement of the PRC2 complex during di-nucleosome binding, which makes sense sterically and is the most parsimonious given the localization of the histone tail-binding regions at opposite ends of the complex. We summarized these ideas in just a single paragraph and describe it in an additional figure (Figure 13) that was extracted from the previous Figure 12 (now simplified), and that lacks the reference to H1.

•*Based on the combined results, the authors make a number of suggestions to provide a structural basis for previous biochemical observations. These include how the enzymatic activity is affected by modifications of the H3 tail, as well as the stabilization of the complex by AEBP1. These inferences about biochemical function are speculative given the resolution of the structure*.

We understand the concerns of the reviewers. And while we do agree that at the resolution of the structure we present here we cannot describe specific mechanistic details concerning single amino acid interactions, which will only come when atomic details are available, our present work still represents a huge step forward that allows, for the first time, to place previous biochemical data into an architectural framework of PRC2. During the past few years, work from different labs have shown how the enzymatic activity of PRC2 is affected by modifications of Histone H3. It has also been shown how the binding of AEBP2 affects and increases its activity. In this paper we describe the first molecular organization of the PRC2 complex and how AEBP2 interacts with different regions of the complex. In the discussion section we are simply placing the published biochemical data in the framework of the structure and protein connections we have visualized. This is something that could not be done before and that opens new questions and testable models. We believe that the localization we describe here does lead to a parsimonious explanation of biochemical data that we truly feel should be indicated at this point. To address the reviewers’ concerns we have revised the text to clarify how far the present data takes us, making clear where we are just placing biochemical data into a physical context, and when we are extrapolating, in a logical way, to propose testable models that really build on our new structural data, always keeping in mind the resolution limitation.

•*We were wondering why only 20 classes were considered for the Random Conical Tilt method. If the complexes were homogeneous this would correspond to an angular difference of 40 or 50 degrees between classes which is a large merging angle and would lead to a rather poor resolution. If several different conformations were to be considered an even larger merging angle would be considered. The authors should justify why only 20 classes were reconstructed*.

We made 20 classes based on the number of tilt pairs we had, since this number could distribute the dataset into classes that had an appropriate number of particles for a reconstruction with enough signal and at a resolution that was sufficient to serve as an initial model for projection matching. Had we made more classes, then the RCT reconstructions would have had much fewer particles, and not been interpretable. As we relied on a good “tomographic” coverage of the views around the long axis of the structure, we were expecting to have a coverage of about 18 degrees from the 20 classes (360/20). We believe this was a good compromise between angular spread and signal. A proof of this principle was that our strategy actually worked, generating an RCT 3D model from which reprojections matched well our reference free 2D classes, and that ultimately aided to refine our untilted data to 21 Å.